# The Evolution of Reverse Total Shoulder Arthroplasty and Its Current Use in the Treatment of Proximal Humerus Fractures in the Older Population

**DOI:** 10.3390/jcm11195832

**Published:** 2022-09-30

**Authors:** Gabriel Larose, Mandeep S. Virk

**Affiliations:** 1Health Sciences Center, Division Orthopedic Trauma, University of Manitoba, Winnipeg, MB R3A 1R9, Canada; 2Division of Shoulder and Elbow Surgery, Department of Orthopedic Surgery, NYU Langone Orthopedic Hospital, 301 East 17th St., New York, NY 10003, USA

**Keywords:** proximal humerus fractures, reverse total shoulder arthroplasty, fragility fractures, hemiarthroplasty, internal fixation

## Abstract

Proximal humerus fracture (PHF) is a common injury in the older population. While the majority of these fractures are treated non-operatively, a small subset of patients may benefit from surgical treatment. However, there continues to be an ongoing debate regarding the indications and ideal surgical treatment strategy. The use of reverse total shoulder arthroplasty (RTSA) has resulted in a paradigm shift in the treatment of PHFs in the older population. Unique biomechanical principles and design features of RTSA make it a suitable treatment option for PHFs in the older population. RTSA has distinct advantages over hemiarthroplasty and internal fixation and provides good pain relief and a reliable and reproducible improvement in functional outcomes. As a result, there has been an exponential increase in the volume of RTSA in the older population in last decade. The aim of this paper is to review the current concepts, outcomes and controversies regarding the use of RTSA for the treatment of PHFs in the older population.

## 1. Introduction

Proximal humerus fracture (PHF) is a common injury in the older population and is the second most common upper extremity fracture after distal radius fracture [1,2]. Low energy PHFs, such as falls from standing height, are often associated with osteoporosis and therefore, are considered fragility fractures [3]. Advancing age and osteoporosis predispose to increasing incidence of PHFs, and therefore it has been projected that there will be an increase in the use of RTSA for these difficult fractures in the future [2,4].

The majority of PHFs are amendable to nonoperative treatment [3] but there continues to be an ongoing debate regarding the ideal treatment of more complex, displaced PHFs. Although recent randomized clinical trials have recommended nonoperative treatment for all displaced 3- and 4-part fractures, there is a subset of patients with these fracture patterns that do benefit from surgical treatment [5,6]. However, there is no consensus on the ideal surgical treatment modality for these fractures. Operative treatment includes internal fixation (locking plates, humeral nails, percutaneous fixation) versus arthroplasty (hemiarthroplasty [HA] and reverse total shoulder arthroplasty [RTSA]) [7].

The use of RTSA in the treatment of PHFs has resulted in a paradigm shift in the surgical treatment since its introduction in the United States in 2004 [1,8,9]. As a result of the increased use there has been a considerable evolution in the role of RTSA in PHF in the last 2 decades. In this narrative review, we present the current concepts, outcomes and controversies regarding the use of RTSA for the treatment of PHFs in the older population (chronological age > 65 years) [10]. 

## 2. Evolution of Arthroplasty for the Treatment of Proximal Humerus Fracture

In 1950’s Charles Neer [11] reported on the unsatisfactory outcomes of 20 patients with PHF treated with closed reduction, humeral head excision, or arthrodesis. Based on his preliminary good results with hemiarthroplasty, he proposed prosthetic replacement of the humeral head as a treatment option for displaced comminuted PHFs. Charles Neer subsequently treated 12 PHFs with hemiarthroplasty and reported improved function in this cohort compared to patients who underwent excision arthroplasty. The results of this study led to introduction and popularity of HA in the treatment of PHFs [12]. However, HA remains a technically challenging surgery and subsequent studies, by other investigators, have shown less successful results compared to those reported by Neer [13,14,15]. The short-term outcomes of hemiarthroplasty are variable and are directly affected by healing of the tuberosities [16]. The mid-term and long-term results of HA are negatively affected by the glenoid chondral wear, prosthetic instability, and pre-existing rotator cuff dysfunction or new rotator cuff tears and these results are unpredictable and often result in an abbreviated life span of the hemiarthroplasty [17]. 

Introduction of RTSA was the next big breakthrough in the treatment of PHFs with arthroplasty (Figure 1). Although initially introduced for the treatment of rotator cuff tear arthropathy in patients >75 years, the indications for RTSA have undergone a considerable evolution and now includes treatment of non-reconstructable PHFs, fracture dislocations and PHF sequela [8,9,18]. The current indications for RTSA in acute and salvage situations for PHF, based on the best available evidence in the literature, are listed in Table 1.

## 3. Rationale for Use of RTSA for the Treatment of PHFs

RTSA is a non-anatomic, semi-constrained shoulder arthroplasty design. The inverse or reverse design includes a metallic baseplate, which is fixed to the glenoid generally via a central post and peripheral screws and to which a glenosphere is attached. The glenosphere articulates with a polyethylene liner that is fixed to the humeral component via a metallic humeral tray (Figure 2). Biomechanically, the center of rotation (COR) in RTSA is fixed and is located on or close to the face of the glenoid as compared to the anatomic shoulder arthroplasty where the COR is located on the humeral head. As a result of the COR being medialized, the lever arm of the deltoid is lengthened increasing the efficiency of this important muscle. Furthermore, the COR is distalized, which allows for an effective tensioning of the deltoid and provides soft tissue stability (Figure 2). The combination of medialized and distalized COR converts the shearing force of deltoid into compressive force across the glenohumeral articulation thereby allowing for forward elevation and abduction even in the absence of a functional rotator cuff. These biomechanical principles and the design features of current RTSA make it a suitable treatment option for PHFs in the older patients. As the rate of tuberosity healing is lower and/or resorption of tuberosities is not uncommon in this patient population, RTSA provides a predictable forward elevation and above shoulder level function irrespective of the status of tuberosity healing and rotator cuff function [14,16]. The presence of healed tuberosities allows for good rotation control of the arm in space and recent studies have shown improved postoperative range of motion and lower risk of instability when tuberosities heal [19]. 

The original Grammont’s version of RTSA has undergone several modifications since its inception offering unique advantages tailored for particular clinical situation and surgeon’s preference (Figure 3). Additionally, components of different sizes and modularity in the stem neck shaft angles, and glenosphere (lateralized, inferior offset) are available (Figure 2A) to adjust soft tissue tension for achieving maximal stability while minimizing the risk of glenoid notching and acromion stress fracture. 

The RTSA, by its design, has several advantages over hemiarthroplasty, which has led to the increased popularity of this implant for treatment of PHFs in the older population. First, RTSA involves prosthetic replacement of the humeral head and glenoid and eliminates the concern for pain resulting from preexisting shoulder arthritis or future glenoid chondral wear, which can be seen with HA. Second, RTSA is a semi-constrained design relying on the deltoid muscle for stability unlike hemiarthroplasty that requires a functioning rotator cuff and healed tuberosities for achieving prosthetic stability. Third, tuberosity healing is less critical for achieving shoulder-level function in RTSA compared to hemiarthroplasty [21,22]. Lastly, the nonanatomic nature of the arthroplasty design and availability of multiple sizing options of the modular components makes soft tissue tensioning easier in RTSA compared to HA. The advantages of RTSA over hemiarthroplasty are particularly attractive in older patients with PHFs because of high prevalence of asymptomatic rotator cuff tears, poor healing of tuberosities and possibility of underlying glenohumeral osteoarthritis. Consequently, RTSA has become the most popular arthroplasty option for the treatment of PHFs in the in older patients [23].

## 4. Current Controversies in the Treatment of Proximal Humerus Fractures in the Elderly

Majority of the proximal humerus fractures in the older patient population can be treated non-operatively with an acceptable functional outcome. A recent epidemiology study showed that in fact 67% of the PHFs in the Medicare Database were treated non-operatively [3]. However, the treatment of displaced 3- and 4-part PHFs in this older population continues to be controversial for two primary reasons. First, there is a subset of patients within the displaced 3- and 4-part cohort, which will benefit from surgical treatment. Unfortunately, radiographic assessment is not the only determinant in the selection of this cohort and the decision-making is complex and involves multiple other factors including patient’s pain scores, preinjury activity level and ambulatory status including the use of weight bearing walking aids, living status (home versus institutionalized), handedness, desired postoperative function, function of lower extremities and spine, functional use of ipsilateral hand, and presence of medical comorbidities [6,24]. The individual contribution of these patient-related factors varies with each patient and treatment algorithms incorporating these factors are currently lacking. Systematic reviews and clinical practice guidelines report insufficient evidence to definitively guide the treatment of these fractures [25,26]. However, in an attempt to develop such guidelines, a pragmatic, multicenter, parallel-group, randomized trial, a study titled PROFHER, was performed [6]. The PROFHER [6] study analyzed 250 patients with an acute (<3 weeks old), displaced surgical neck of the humerus fracture treated at 33 centers. The investigators concluded, at 2 years post-operatively, there was no significant differences in outcomes (Oxford Shoulder score) between surgical and non-surgical treatment. Unfortunately, the results of this study are disputed in part because of selection bias; exclusion of patients with dislocations and subset of patients with “clear indication for surgery”. Although multicenter involvement is a strength for any study, conversely it also introduces inconsistencies in treatment selection (nail vs. plate vs. hemiarthroplasty vs. reverse total shoulder replacement), which has been shown to affect outcomes [27,28,29]. Nevertheless, the goal was to assess the patients with equivocal treatment, and they showed no significant differences in outcomes. A follow up study at 5 years, showed the same results [5]. Second, there is no consensus on the ideal treatment method for displaced fractures in the older patient population. Presence of osteoporosis, poor bone stock and underlying preexisting osteoarthritis or rotator cuff tears, and duration of fracture and associated dislocation are some of the factors that prevent selection of universal gold standard surgical treatment of PHFs in the older population.

## 5. Evidence for the Use of RTSA in PHF Compared to Other Surgical Treatments

### 5.1. RTSA versus Nonoperative Treatment

Since most PHF are amenable to nonoperative treatment, there is paucity of studies directly comparing nonoperative and RTSA [3]. In a randomized control trial, Lopiz et al. [30] randomized 62 patients over the age of 80 with 3- or 4-part PHF to either RTSA or nonoperative treatment. Patient with fracture dislocation and head split were excluded in the study. At the 1-year follow up, the results favored RTSA over the non-operative groups with respect to VAS score (0.9 vs. 1.6; *p* = 0.01), Constant score (55.7 vs. 61.7; *p* = 0.07) and DASH score (28.8 vs. 20.7; *p* = 0.08). Furthermore, a higher forward flexion was observed in the RTSA group (Constant ROM: 5.7 vs. 6.9 *p* = 0.03) versus non-operative treatment but no differences was observed with respect to external rotation (*p* = 0.29) and internal rotation (*p* = 0.21). Two patients in the RTSA group had postoperative suprascapular nerve injury. In a retrospective study, Chivot [31] compared nonoperative treatment with RTSA in patients with PHF (>70 years of age). They reported an improved Constant score in the RTSA group compared to nonoperative treatment (56.5 vs. 50.5; *p* = 0.03 value, MCID 5.7 [32]) but there was no difference in the DASH score. They also reported an improved range of motion in the RTSA group (forward flexion 110° vs. 98° *p* < 0.01; external rotation 19° vs. 9° *p* < 0.01 and internal rotation *p* = 0.04). Moreover, in this study, patients who underwent a RTSA reported a higher satisfaction than patient with non-operative treatment. Another smaller retrospective study [33] comparing patients greater than 50 years (mean age 71, range 72–88), reported no differences in the ASES score, VAS pain scores and range of motion. It is important to point out that these studies typically do not include fracture dislocations or head split fractures, which are absolute indications for operative treatment, because non-operative treatment has been associated with poor outcomes. Furthermore, all 3- and 4-part fractures are not the same and the challenge of patients who cannot tolerate nonsurgical treatment are difficult to define objectively and tend to dilute the advantage of RTSA in these studies. There is weak evidence to support for RTSA over nonsurgical treatment in elderly patients with 3- or 4-part proximal humerus fractures in absence of fracture-dislocation or head split fractures (grade C recommendation) [34]. 

### 5.2. RTSA versus Internal Fixation

The advent of locking plates has made the treatment of osteoporotic PHFs more amenable to surgical intervention and has reported improved functional outcomes [27,35,36,37,38,39,40,41,42]. However, maintaining an anatomic reduction and avoiding osteonecrosis, screw penetration, malunion and nonunion continues to be a challenge with the use of locking plates [43]. Unlike internal fixation, outcomes after RTSA do not rely on the vascularity of the humeral head (post-injury or post-surgery), or bone quality of the humeral head for screw purchase. Additionally, RTSA can be more easily performed in a delayed setting and are less affected by the lack of tuberosity healing compared to when internal fixation is performed.

Fraser [27] compared RTSA and ORIF in a randomized control trial (RCT) of patients with PHF (65–85 years of age). One-hundred and twenty-four patients with PHF AO/OTA B2 and C2 were included in the trial. Patients with fracture dislocations, head split, high-energy trauma, and with other injuries, were excluded. At 2 years, compared to the ORIF group, patients in the RTSA group demonstrated improved Constant score (RTSA 68 vs. ORIF 54.6; *p* < 0.001), which was attributable to improved ROM and strength (forward flexion: 7.0 vs. 5.2; abduction 6.4 vs. 4.6; external rotation 7.0 vs. 4.4 strength 11.8 vs. 8.8 *p* < 0.05). There were 7 adverse events in the RTSA group (2 transient nerve injuries, 2 deep wound infections, 2 periprosthetic fractures, and 1 perioperative glenoid fracture) with 4 re-operations (2 with component exchanges and 2 with other revisions not specified). There were 12 adverse events (9 screw penetrations, one periprosthetic fracture, 1 non-union, 1 rotator cuff tear) in the ORIF group with 8 re-operations (4 conversion to a RTSA, 4 hardware removal). This study suggests that RTSA for treatment of displaced PHFs in the older patient results in improved functional outcome, compared to the ORIF, and also illustrates the risks and complications of both surgeries. 

Multiples retrospective studies have compared RTSA to ORIF in the treatment of PHF (Table 2), the outcomes vary amongst these different studies [35,36,37,38,39,40,41,42]. When assessing patient reported outcomes, no significant differences were seen between RTSA and ORIF but RTSA demonstrated better ranges of motion, especially in forward flexion [35,36,37]. Additionally, these studies show a trend towards more complications and more revision surgeries in the ORIF group [37,38,39,40,42]. However, 3- or 4-part fracture dislocations and head split fractures which are typically treated with arthroplasty were excluded from most of these studies.

Although RTSA demonstrates better outcomes compared to ORIF in the elderly patient population, more evidence is necessary to demonstrate superiority of RTSA over ORIF for displaced 3- or 4-part PHFs in the patients >65 years of age. 

### 5.3. RTSA versus Hemiarthroplasty

Prior to the availability of RTSA, hemiarthroplasty was a treatment option for non-reconstructable, displaced PHFs and fracture-dislocations. The RTSA has surpassed hemiarthroplasty as the treatment of choice in PHFs that require treatment with arthroplasty. Multiple studies have compared RTSA and the HA for the treatment of acute PHFs. In a RCT [29] comparing HA to RTSA for the treatment of acute PHFs in patient greater than 70 years. Sebastia-Forcada et al. demonstrated significantly better (*p* = 0.001) Constant and DASH scores in the RTSA group compared to the HA group. The RTSA group also had a significantly higher (*p* = 0.001) forward flexion, abduction, and external rotation and a lower revision rate compared to the HA group. There was no difference in the rate of tuberosity healing between the two groups. Six patients in the HA required revision to RTSA secondary to rotator cuff insufficiency. At 40 months, RTSA had significantly higher survival rate (*p* < 0.05) compared to HA considering revision for any cause and/or clinical failure as the end point. Findings similar to the RCT by Sebastia-Forcada et al. were reported in a recent RCT by Jonsson et al. [28] evaluating in patients greater than 70 years. At a mean follow up of 2.4 years, they found that patients treated with RTSA demonstrated better Constant score (primary outcome measure), range of motion (except for internal rotation) and satisfaction score compared to HA. There were no differences between the two groups with respect to WOOS index score, EQ-5D index score, and tuberosity healing. Another recent RCT [44] comparing RTSA to HA for acute PHF (*n* = 33) in patients greater than 65 years demonstrated improved DASH score and range of motion (forward flexion and abduction) with RTSA. There are multiple non-randomized prospective and retrospective studies [35,41,45,46,47,48,49,50,51] comparing RTSA and HA in the treatment of PHF (Table 3). Although there are some discrepancies in the results between the studies, overall RTSA had an improved ASES, Constant score, and ROM compared to HA.

Based on the available evidence, RTSA is preferred to HA for displaced 3- or 4-part PHFs and fracture-dislocations in patients >65 years of age (Grade B recommendation) [34].

### 5.4. Cost Effectiveness Analysis

Information on the cost effectiveness of each treatment in PHF is sparse [52,53,54,55]. In a recent study from Norway, Bjørdal et al. [53] compared the cost-effectiveness of RTSA with ORIF in elderly patients (65–85 years old) with PHF. In their study, they found that the incremental cost of RTSA, compared to ORIF, was EUR 4802. The incremental effect was 0.02 QALYs in favor of ORIF. In contrast to the study by Bjordal et al., other studies have found RTSA to be a cost-effective strategy for PHFs in elderly population. Khalik et al. [55] found that at a willingness to pay threshold of $50,000/QALY (quality-adjusted life years) the RTSA had 66% probability of being the most cost-effective treatment compared to hemiarthroplasty, ORIF or non-operative treatment. Using a base case of 75-year old with PHF, Austin et al. demonstrated that RTSA was associated with greater quality of life (7.11 QALYs) than ORIF (6.22 QALYs). RTSA was cost-effective with an ICER (incremental cost-effectiveness ratio) of $3945/QALY and $27,299/ QALY from payor and hospital perspectives, respectively. RTSA was favored and cost-effective at any age above 65. Nwachukwu et al. [54] compared cost effectiveness of RTSA, HA and nonoperative treatment using a base case of 70 year old with a complex PHF. They showed that ICER for RTSA was $8100/QALY and ICER for HA was $36,700/QALY compared with nonoperative treatment. At a cost-effectiveness threshold of $100,000/QALY, RTSA, HA, and nonoperative intervention were, respectively, the optimal cost-effective strategies in 61.0%, 34.7%, and 4.3% of payor analyses and 53.8%, 37.4%, and 8.8% of hospital analyses. 

## 6. Complications of RTSA

There is a concern about the higher risk of certain complications in RTSA for the treatment of PHF compared to RTSA performed for rotator cuff tear arthropathy [56,57,58]. The risk of hematoma formation (secondary to bleeding from trauma and fracture), prosthetic instability (failure of tuberosity healing) and periprosthetic fracture (intraoperative or postoperative); secondary to underlying osteoporosis are postulated to be more pronounced with the use of RTSA for PHFs [56,57,58,59]. A recent systemic review reported an overall 11% risk of complication with RTSA performed for treatment of acute PHFs with a higher risk of prosthetic instability and periprosthetic fractures but lower risk of acromial stress fracture compared to RTSA done for cuff tear arthropathy [57]. 

## 7. Special Considerations for RTSA for Fractures

### 7.1. Press Fit versus Cemented Humeral Stems

Although RTSA with cemented humeral stems have been traditionally used in acute PHFs but uncemented humeral stem designs have become increasingly available and are being more widely used. The cemented stem offers immediate rotational and vertical stability in the context of metaphyseal bone loss particularly in the setting of underlying osteoporosis. The uncemented, press fit stems on the other hand have theoretical advantages of providing biological fixation including tuberosity healing, lower risk of cement related complications (nerve injury, marrow-embolism), shorter operative time and ease of future revision. However, uncemented stems carry a theoretical risk of early subsidence or loosening due to lack of rotational stability if the press fit stability is lost in the early postoperative period. The role of uncemented, press fit stem in RTSA for PHFs has been reported in several studies but only a few of them directly compare uncemented to cemented humeral stems [60,61,62,63,64,65]. Lopiz et al. [61] retrospectively reviewed 68 patients with primary RTSA for PHF, which included 45 cemented and 23 press fit humeral stems. Although, cemented stems had lower stress shielding (0% vs. 30.4%; *p* < 0.01) and higher forward flexion (127° vs. 108°; *p* = 0.03), uncemented stems had higher anatomic tuberosity healing (*p* = 0.02) and low risk of radiographic loosening. Rossi et al. [62] retrospectively reviewed and compared cemented and uncemented humeral stem RTSAs in 67 patients (32 cemented and 35 press fit humeral stems) with acute PHFs at a mean follow up of 41 months. They found no differences between the two groups with respect to ASES score, Constant score, VAS pain, ROM, tuberosity healing, and complication. There was no aseptic loosening of the humeral stem in the uncemented group. Clinical and radiographic outcomes similar to Rossi et al. were reported by Schoch et al. [63] in 38 patients who underwent RTSA for PHF (19 cemented, 19 uncemented). Although they reported better ASES (*p* = 0.005) and satisfaction scores (*p* = 0.04) in the cemented group, the two groups were similar with respect to VAS pain, range of motion, tuberosity healing, complication, and revision rates. A recent systematic review reported no differences between the uncemented and cemented humeral stems with respect to the Constant score, VAS pain scores, ROM, tuberosity healing and reoperation rates at a mean follow up of 34.6 months. However, uncemented stems were associated with a higher rate of complications (9.7% vs. 5.5%; *p* = 0.044) compared to cemented stem cohort [62]. Although it is too early to conclude if one stem option is superior to the other in the setting of PHF, available short-term data demonstrates comparable functional results of RTSA with uncemented stem compared to cement humeral stem for PHFs.

### 7.2. Standard Humeral Stem versus Fracture Stem 

With the introduction of RTSA for the treatment of PHF humeral stem designs were modified to allow for easier tuberosity repair and facilitate healing of tuberosities (Figure 4). Some of the design features include presence of holes in the lateral fin of the humeral component (Figure 2) to allows for secure tuberosity repair, as well as a “metaphyseal window” proximally to allow for bone growth across the tuberosities and a coated metaphyseal stem to promote tuberosity and shaft osseous integration and healing [66]. Although there is some evidence that fracture stem is associated with higher tuberosity healing in HA, there is insufficient evidence in literature supporting superiority of a fracture stem over standard stem in RTSA [67,68]. Multiple retrospective studies and systematic reviews report conflicting evidence with respect to tuberosity healing and or shoulder function when using a fracture stem compared to a standard humeral stem RTSA for PHFs [66,68,69,70,71,72,73,74]. 

### 7.3. Tuberosity Healing and Its Influence on Outcomes of RTSA

RTSA was originally introduced to treat rotator cuff deficient shoulders (rotator cuff tear arthropathy) and therefore it has been postulated that healing of the tuberosity in PHF may not matter and is less critical for achieving optimal post-operative function. Therefore, some authors reported excising the tuberosities as part of the RTSA in the treatment of PHF [59,75,76]. However, it is important to note that PHF is an acute condition and adaptive mechanisms to combat acute rotator cuff deficiency as compared to patients with rotator cuff tear arthropathy, which generally occurs gradually over several years. It has been postulated that greater tuberosity healing will improve external rotation and joint stability and therefore excision of tuberosity should be avoided. Gallinet et al. [77] demonstrated that patients with tuberosity repair did better than patient with no repair or tuberosity resection, even when the tuberosities did not heal. Tuberosity healing rate with RTSA has been reported to be approximately 70% (range: 37–100%) [78,79,80]. There is conflicting evidence regarding humeral stem inclination (135° vs. 145° vs. 155°) and lateralization of RTSA affecting tuberosity healing and overall functional outcome [80,81]. Numerous fixation techniques have been described to improve healing of the tuberosities in RTSA, but there is no consensus on a single gold standard technique [22,66,78,81,82,83,84,85,86]. There is Level 2 and 3 evidence to demonstrate that tuberosity healing is associated with improved function and range of motion and low risk of humeral loosening and instability [22,62,77,87]. Therefore, it appears to be advantageous to repair the tuberosities in an effort to maximize the chances of an anatomic healing and improve shoulder function and prosthetic longevity. 

### 7.4. Timing of RTSA

Unlike other surgical treatments for PHFs, RTSA has the advantage that surgery can be delayed for few weeks without affecting the outcomes. Therefore, an initial trial of nonoperative treatment for displaced and comminuted PHF is a plausible strategy in patients who are not medically optimized or unsure about proceeding with surgery [88]. Currently, there is inconclusive evidence to demonstrate meaningful difference between early and delayed RTSA for acute PHFs. Considerable healing of the tuberosities is expected in first 6–8 weeks and therefore studies have reported time ranging from 3 to 6 weeks as the cutoff between early and delayed RTSA for acute PHFs [89,90,91,92]. Barger et al. demonstrated that delaying RTSA beyond 4 weeks after PHF did not increase the complication rate but was associated with lower DASH scores (*p* = 0.034). However, Seidel et al. demonstrated that delaying RTSA for >4 weeks, in the setting of an acute PHF is associated with higher short-term revision rate and dislocation rate compared to earlier intervention (<4 weeks) [91]. Schwartz et al. [93] compared the outcomes of RTSA in an acute setting versus RTSA for fracture sequela or salvage situation in a retrospective study with a minimum follow up of 2 years and minimum age of 60 years. They compared the outcomes of 42 RTSA done for acute PHF (<10 days) with 26 RTSAs done for delayed fracture care or revision indication (10 failed ORIF, 12 failed non-operative treatment, and 4 failed HA). In this study, no significant differences were found for outcome scores between the groups. However, subgroup analysis showed that failed osteosynthesis group was associated with lower ROM (abduction and flexion). 

### 7.5. RTSA as a Salvage Procedure

RTSA is indicated as a salvage operation for failed internal fixation of PHFs. The main cause of failure of ORIF after PHFs is screw cut out with fracture displacement, humeral head osteonecrosis and glenoid wear [94]. Several retrospective cohort studies have shown improved PROM scores and ROM outcomes after a salvage with a RTSA [94,95,96]. Furthermore, primary RTSA in an acute setting is associated with slightly better PROM, ROM and less complication rate compared to RTSA after failed osteosynthesis [93,97,98,99,100]. 

## 8. Conclusions

Although the majority of the PHFs can expect to have a good functional outcome with nonoperative treatment, there is a small subset of patients with displaced fracture (3- and 4-part fractures), and patients with complex fracture-dislocation and head split fractures that will benefit from surgical intervention. The increasing success of RTSA for the treatment of PHFs has resulted in paradigm shift in the management of these injuries, particularly in older patients. As a result, there has been an exponential increase in the volume of RTSA in the last decade. Unique biomechanical principles and design features of RTSA make it a suitable treatment option for PHFs in the older population. RTSA has distinct advantages over hemiarthroplasty and internal fixation and provides good pain relief and a reliable and reproducible improvement in functional outcomes. Future research is necessary to better define the indications for surgery and RTSA in PHFs to optimize the outcomes, increase longevity of the implants, while reducing complications.

## Figures and Tables

**Figure 1 jcm-11-05832-f001:**
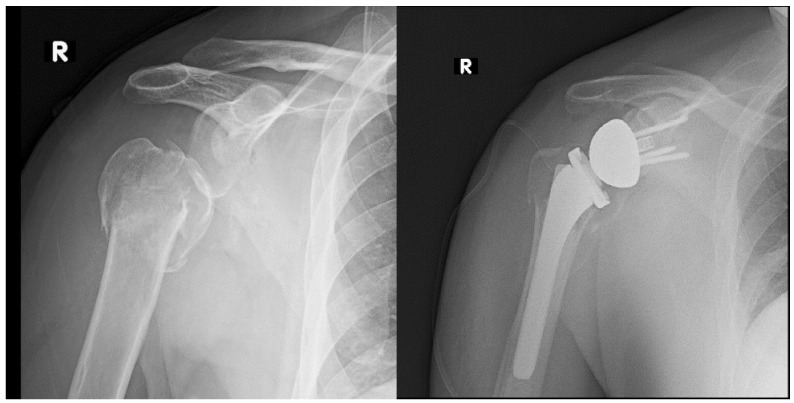
Preoperative and postoperative radiographs of Reverse Total Shoulder Arthroplasty (RTSA) with tuberosity reconstruction for a displaced, comminuted 4-part impacted proximal humerus fracture (PHF). R—radiographic marker for the right side.

**Figure 2 jcm-11-05832-f002:**
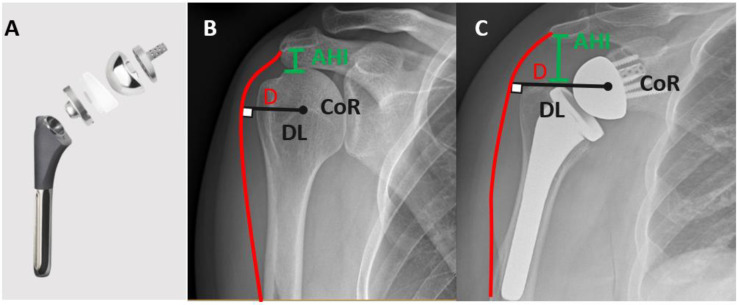
Reverse total shoulder implant (**A**) Biomechanics of a native shoulder (**B**) compared to a reverse total shoulder (**C**); Deltoid lever arm (DL), acromiohumeral interval (AHI) and location of center of rotation (CoR), Deltoid muscle (D).

**Figure 3 jcm-11-05832-f003:**
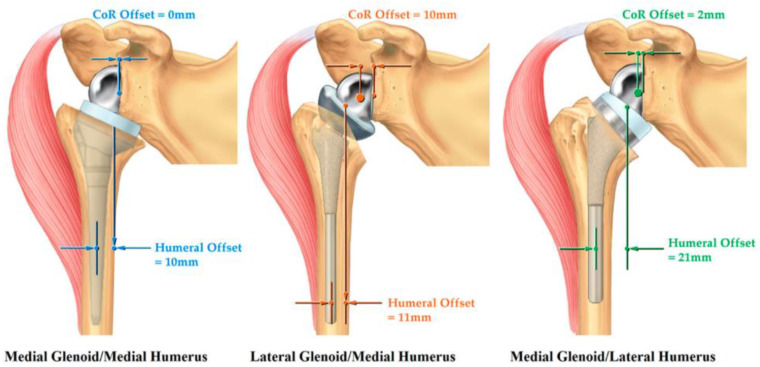
Classification system for RTSA described by Routman et al. From left to right: Medial glenoid/Medial Humerus design (Original Grammont design); Lateral Glenoid/Medial Humerus design (inlay humerus with lateralized glenoid); Medial Glenoid/Lateral Humerus (Onlay humerus with medialized glenoid). Lateralized humerus design (>15 mm humeral offset); Lateralized glenoid design (>5 mm COR offset) COR: center of rotation [20].

**Figure 4 jcm-11-05832-f004:**
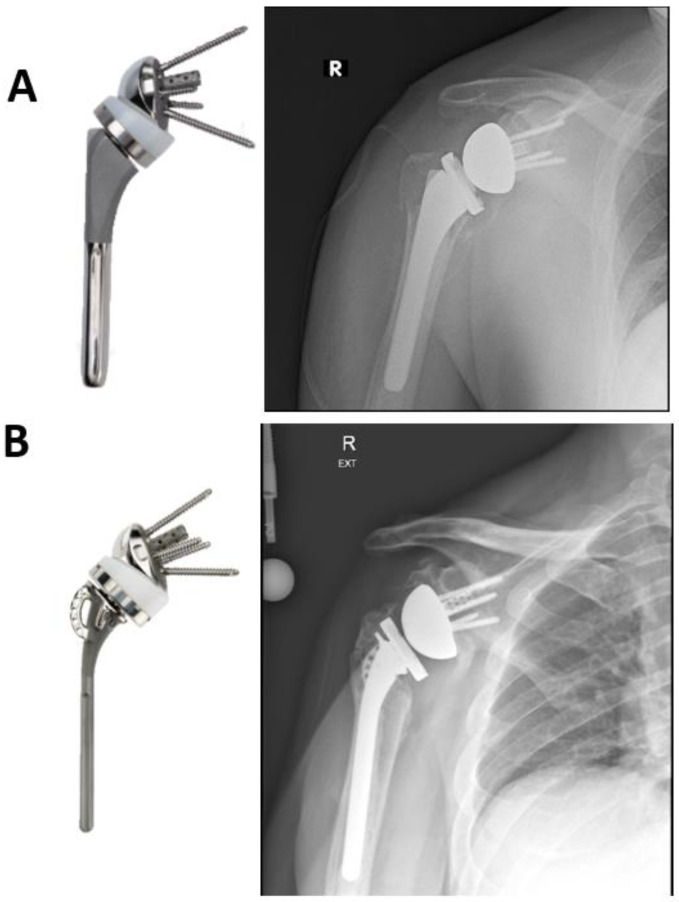
Conventional (**A**) and fracture specific (**B**) humeral stem.

**Table 1 jcm-11-05832-t001:** Surgical Indications.

**Acute Fracture**
Displaced 3- and 4-part proximal humerus fracture Displaced Head split fractures 3 or 4-part proximal humerus fracture-dislocation
**Salvage Indications**
Failed hemiarthroplasty- acute or chronic Failed internal fixation (plates, nail, percutaneous fixation) Fracture sequela (malunited proximal humerus, nonunion)

**Table 2 jcm-11-05832-t002:** **Studies** comparing RTSA vs. ORIF for the treatment of proximal humerus fractures.

Study	LOE	Patients	Age	PROMS	ROM FF	ROM ER	Complications	Revisions
Chalmers	III	ORIF: 9	71	ASES: 75	108	46	Stiffness1	1
2014 [35]		RTSA: 9	77	ASES: 80	133	41	CRPS	0
Giardella	III	ORIF: 23	72.1	CMS 52.9	112.8	47.4 *	NR	NR
2017 [36]		RTSA: 21	77.2	CMS 65.9 *	133.3 *	35.5	NR	NR
Greiwe	III	ORIF 25	73.3	ASES: 81.1	121.4	43	AVN:4; Screw cut out 2; Nerve palsy1; Delayed union 1; malunion 2	6 *
2020 [37]		RTSA: 25	74.4	ASES: 82.9	143.2 *	46.8	Tuberosity resorption 5	0
Klug	III	ORIF: 66		NR	NR	NR	Stiffness 17, AVN: 6; loss of fixation 4; screw cut out 2; infection 1 PE 2; anemia 1	7
2019 [39]		RTSA: 59		NR	NR	NR	Stiffness 9; Instability 3; Axial nerve palsy 2; radiolucent line 2; 2; PE 2; anemia 1	3
Klug	III	ORIF 30	72.5	ASES: 83.4 CMS 81.4 DASH 14.3 *	146	52	Stiffness6; loss of fixation 2; screw cut out 1; infection1	6
2020 [38]		RTSA 30	73.9	ASES: 74.6 CMS 69.9 DASH 25.3	133	39	Axillary nerve1; dislocation 1; infection 1	1
Luciani	III	ORIF: 26	73	CMS 65.85 DASH 18.99	125.75	28 *	AVN5; loss reduction3; infection1; hardware impingement 2;	7
2022 [40]		RTSA: 22	75.5	CMS 63.65 DASH 25.1	124.5	14.25	Instability1; infection1	2
Repetto	III	ORIF: 19	65.3	CMS 61.8 DASH 16.9	130.6	23.2	AVN:4; Hardware impingement: 2; Transient circumflex nerve palsy 1	3
2017 [41]		RTSA: 27	71.2	CMS 58.5 DASH 28.6	125	20.3	Infection: 1; Hematoma: 1; Periprosthetic fracture: 1; Instability 2;	3
Yahuaca	III	ORIF 211	61.6	NR	130		Tuberosity nonunion 22 *	17.50%
2020 [42]		RTSA: 106	73	NR	124		Tuberosity nonunion 25	6.6% *

* *p* < 0.05. LOE: Level of Evidence; ASES American shoulder and elbow society score; CMS: Constant-Murley score; DASH: Disabilities of the arm, shoulder and hand score; NR: not reported; ORIF: open reduction internal fixation; RTSA: reverse total shoulder arthroplasty.

**Table 3 jcm-11-05832-t003:** **Studies** Comparing RTSA vs. HA for the treatment of Proximal Humerus Fractures.

Study	LOE	Patients	Age	PROM	ROM FF	ROM ER	Tub Healing	Complications	Revisions
Baudi	III	RTSA: 25	77 *	CMS 56.2 * DASH 40.4	131*	15	84% *	1 transient nerve palsy	NR
2014 [45]		HA: 28	70	CMS 42.3 DASH 46.1	89	23	27%	2 septic infections; 1 Pulmonary Embolism; 3 Stiffness	NR
Bonnevialle	III	RTSA: 41	78 *	CMS 57 DASH 28	130 *	23	73%	1 hematoma; 1 transient nerve injury; 2 HO	0
2016 [46]		HA: 57	67	CMS 54 DASH 30	112	28	72%	11 stiffness; 1 HO; 1 infection; 1 transient nerve palsy	1
Chalmers	III	RTSA: 9	77	ASES 80	133 *	41	100%	1 Complex Regional pain Syndrome	0
2014 [35]		HA: 9	72	ASES 66	106	28	100%	1 Ulnar nerve neuritis; 1 Stiffness	0
Cuff	III	RTSA: 24	NR	ASES 77 *	139 *	24	67%	8 complications -not specify	0
2013 [48]		HA: 23	NR	ASES 62	100	25	57%	9complications -not specify	3
Garrigues	III	RTSA:10	80.5 *	ASES 81.1	121 *	34	100%	none	0
2012 [49]		HA:9	69.3	ASES 37.4	91	31	22%	2 transient nerve palsy; 1 periprosthetic fracture; 1 glenoid erosion	3
Repetto	III	RTSA: 27	71.2	CMS 58.5 DASH 33.8	125	20.3	NR	1 Cuff Failure; 2 Periprothetic fracture; 2 Stiffness	3
2017 [41]		HA: 24	67.5	CMS 48.4 DASH 28.6	103	16.5	79%	2 Instability; 1 Periprosthetic fractures; 1 Hematoma; 1 Deep Infection	7
Young	III	RTSA: 10	77.2	ASES 65	115	49	90%	0	0
2010 [51]		HA: 10	75.5	ASES 67	108	48	80%	1 stiffness; 1 infection	2

* *p* < 0.05. LOE: Level of Evidence; ASES: American shoulder and elbow society score; CMS: Constant-Murley score; DASH: Disabilities of the arm, shoulder and hand score; NR: not reported; HA: Hemiarthroplasty; RTSA: reverse total shoulder arthroplasty.

## Data Availability

Not applicable.

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
