# Peer review of "The Evolution of Reverse Total Shoulder Arthroplasty and Its Current Use in the Treatment of Proximal Humerus Fractures in the Older Population"

_jcm, 2022, doi:10.3390/jcm11195832_

Round 1
Reviewer 1 Report
Dear authors,
congratulations to your nice work. Reading your article was a pleasure and we agree with your conclusions. However, some aspects should be revised or modified:
Clarify the technique of your review. It is a narrative review without a systmatic literature research, or?
Page 4, line 116: delete "is"
Please re-check figure 3. I´m not sure if the middle picture is not correct labeled. Doesn´t it demonstrate a lateral glenoid/lateral humerus instead of medial humerus? In my opinion the different degrees (155-135°) are also important to acknowledge. The figure needs more detailed explanation.
Finally, it is important to mention that RTSA is only justified in a small population.
Kind regards and thank´s for your work.
Author Response
Please see the attachement

Reviewer 2 Report
Thank you for submitting your work. The authors reviewed the articles for current concept and trend in treating proximal humerus fractures using reverse total shoulder arthroplasty (RTSA). They showed the outcome comparison to other treatment options such as non-operative, ORIF, and hemiarthroplasty. Even they required more supporting evidence, RTSA showed more appropriate treatment option for specific indication, such as 3-,4- part fractures, complex fracture-dislocation, and head split fractures. In addition, they summarized the related issues applying RTSA for RSA, such as greater tuberosity healing, cemented, and fracture stem design.
Overall, this is well organized, informative, and written article. Despite some grammatical and expressional errors for requiring English editing process, this review provides interesting updates of recent literatures.
Reviewer 3 Report
The authors have presented a well-done analysis for proximal humerus fracture. However, although the authors state in the text that there is no convincing evidence for the benefit of RTSA (only one ), they argue otherwise in the discussion and summary. A bit more restraint in the argumentation would be appropriate.
In addition, the text does not say a word about the more secure anchorage with angular stable plate osteosynthesis using cement augmentation or the addition of an anterior supplemental plate. The problem is that a trauma surgeon's knowledge and skills are more focused on reconstruction and head preservation, and an orthopedic surgeon's are more flirtatious with arthroplasty because osteosynthesis skills are not as off the top of his or her head. However, this point is missed from the literature listed.
The budgetary aspect of better reimbursement for RTSA, which certainly plays a role in some countries, is also not discussed.
Author Response
Please see attachement
